# Is 'distinctiveness centrality' actually distinctive? A comment on Fronzetti Colladon and Naldi (2020)

**Zachary P. Neal** [ID]*

Psychology Department, Michigan State University, East Lansing, MI, United States of America

* zpneal@msu.edu

**Data Availability Statement:** "The data relevant to this study are available from OSF at https://osf.io/s6pqg/."

**Funding:** The author(s) received no specific funding for this work.

## Abstract

Distinctiveness centrality, which was proposed in 2020 to identify nodes that are connected to poorly-connected neighbors, is simply a minor variation on two existing centrality measures: beta centrality proposed in 1987, and gamma centrality proposed in 2011. In toy, empirical, and generated networks, I show that these three centrality measures yield identical node rankings under nearly all circumstances. Researchers seeking to identify nodes that are connected to poorly-connected others should not use distinctiveness centrality, and instead should use either beta or gamma centrality because they are more widely-known in the literature, are more flexible, and are computationally simpler. Additionally, researchers should be cautious when proposing new centrality measures, taking care to avoid duplicating measures that already exist.

## Introduction

Many measures of network centrality assign higher scores to nodes with many neighbors (i.e., are well-connected), and to those whose neighbors have many neighbors (i.e., are connected to well-connected neighbors). Guided by the belief that sometimes it is advantageous to be connected to *poorly*-connected others, and the belief that existing measures of network centrality can not capture such a position, Fronzetti Colladon and Naldi [1] proposed a family of network centrality measures they called 'distinctiveness centrality.' Through a series of comparisons in toy, empirical, and generated networks, they demonstrated that their new measures yielded distinctive node rankings from degree, closeness, betweenness, and eigenvector centrality [2, 3]. From these analyses, they concluded that "Distinctiveness centrality provides a viewpoint of centrality alternative to that of established metrics," and thereby contributes to the already-vast literature on centrality indices and their relationships to each other [4, 5].

In this brief paper, I demonstrate that distinctiveness centrality is actually not distinctive from two existing centrality metrics: beta centrality originally proposed in 1987 [6] and the computationally simpler gamma centrality originally proposed in 2011 [7]. A series of comparisons in the same toy, empirical, and generated networks reveals that beta and gamma centrality yield identical node rankings under nearly all circumstances. Therefore, distinctiveness centrality should not be regarded as a new centrality measure, but instead as simply a re-

**Competing interests:** The authors have declared
that no competing interests exist.

discovery of existing measures. Researchers seeking to identify nodes that are connected to poorly-connected others should use either beta or gamma centrality because they are more widely-known in the literature, are more flexible, and are computationally simpler. Additionally, researchers should be cautious when proposing new centrality measures, taking care to avoid duplicating measures that already exist.

## Background

### Centrality vs. power

One common goal in network analysis is to score or rank nodes based on the whether their network position is advantageous. Doing so requires specifying *advantageous for what*? Different metrics identify nodes whose positions are advantageous for different outcomes [5]. For example, in the context of a communication network, closeness centrality identifies nodes whose position is advantageous for disseminating a message quickly, while betweenness centrality identifies nodes whose position is advantageous for intercepting a message.

In the context of seeking to maximize one's resources, degree centrality simply counts a node's number of neighbors, which are each a potential source of resources. For example, in a social network, a high-degree person might be advantaged because they have many friends, each of whom is a potential source of social support. Although neighbors are potential sources of resources, some neighbors may be better sources than others. Following this logic, eigenvector centrality extended degree centrality to identify nodes that are connected to well-connected neighbors [3]. Returning to the social network example, a high-eigenvector person is advantaged because they have 'friends in high places' who are potential sources of more or better resources.

However, there are also cases where being connected to well-connected others is not advantageous. Specifically, if one's goal is to exploit one's neighbors, then having well-connected neighbors is a liability. Exploiting poorly-connected neighbors is easier because they have fewer alternative sources for resources themselves (i.e., they are dependent). Indeed, exchange experiments have demonstrated that traders who are connected to well-connected others generate less profit than those who are connected to poorly-connected others [8]. Accordingly, such a position is often described not as a position of 'centrality,' but instead as a position of 'power.'

### Distinctiveness centrality

Distinctiveness centrality aims to identify nodes that occupy positions of power, that is, whose neighbors are poorly-connected [1]. Although distinctiveness centrality was proposed as a set of five closely-related measures, here I focus on 'weighted distinctiveness centrality' (D1), because it is the root measure from which the others were derived, and because despite the name can be computed in both weighted and unweighted networks.

The weighted distinctiveness centrality of node $i$ is defined as

$$D1(i) = \sum_{j,j \neq i} w_{ij} \; log_{10} \; \frac{n-1}{g_j^\alpha}, \tag{1}$$

where $w_{ij}$ is the weight of the edge between nodes $i$ and $j$, $n$ is the number of nodes in the network, $g_j$ is the degree of $j$, and $\alpha$ is a tuning parameter. Like degree centrality, D1 sums the weight of a node's connections, but penalizes connections to highly connected neighbors. The $\alpha$ tuning parameter adjusts the severity of this penalty. The computation of D1 has been implemented in packages available for both Python [9] and R [10].

Fronzetti Colladon and Naldi imply that the allowable range of the tuning parameter is $1 \leq \alpha < \infty$, but do not offer an explanation. In their analyses, they examine the performance of D1 for $1 \leq \alpha \leq 15$, but also do not explain why they chose these values [1]. In the analyses below, I instead investigate $-1 < \alpha < 1$ for two reasons. First, as I show below, D1 has a meaningful interpretation when $\alpha < 1$. Second, it is unclear what very large values of $\alpha$ would mean, or how one would select a value of $\alpha$ from such a wide range.

In their original analysis, Fronzetti Colladon and Naldi compared D1 to four classic measures of centrality: degree, closeness, betweenness, eigenvector. It is unclear why they chose these four for comparison. None of these are intended to identify nodes that are connected to poorly-connected neighbors, and therefore there is no reason to expect that they would be similar D1 [2, 3]. In the analyses below, I instead compare D1 to beta [6] and gamma [11] centrality, which do identify nodes that are connected to poorly-connected neighbors, and therefore are similar to D1.

## Beta centrality

When exchange experiments demonstrated, unexpectedly, that highly central nodes do not generate the most profit [8], 'beta centrality' (BC) was proposed as a centrality measure that could identify advantageous positions in exchange networks [6]. A vector of beta centrality scores are defined as:

$$BC = inv(\mathbf{I} - \beta\mathbf{A})\mathbf{A1}, \tag{2}$$

where $\mathbf{A}$ is an adjacency matrix, $\mathbf{I}$ is an identity matrix, $\mathbf{1}$ is a column vector of 1s, $inv$ is the matrix inverse function, and $\beta$ is a tuning parameter.

The tuning parameter can range $-\frac{1}{\lambda_1} < \beta < \frac{1}{\lambda_1}$, where $\lambda_1$ is the largest eigenvalue of $\mathbf{A}$. When $\beta = 0$, BC is equal to degree. When $\beta > 0$, BC assigns higher scores to nodes that are connected to well-connected neighbors, and converges on eigenvector centrality as $\beta$ approaches $\frac{1}{\lambda_1}$ from below. Finally, when $\beta < 0$, BC assigns higher scores to nodes that are connected to poorly-connected neighbors. Accordingly, BC is conceptually similar to D1 when $\beta < 0$.

Computing BC does not require a specialized software package. In R, it can be computed using:

```
B  <- 0 #Set value of beta
I  <- diag(nrow(A)) #Define identity matrix
O  <- matrix(1, nrow = nrow(A)) #Define vector of ones
BC <- solve(I - (B * A)) %*% A %*% O #Compute
```

## Gamma centrality

Beta centrality's computation can be opaque and the allowable range of $\beta$ can be a source of confusion for users. Gamma centrality aims to overcome these challenges by offering a simplified alternative. This alternative to beta centrality evolved in a series of initial papers in the urban studies [7], psychology [12], and network [13] literatures, but here I focus on its final form, known as 'gamma centrality' (GC) [11]. A vector of gamma centrality scores are defined

as:

$$GC = \mathbf{A}(\mathbf{A1})^{\gamma}, \tag{3}$$

where $\mathbf{A}$ is an adjacency matrix, $\mathbf{1}$ is a column vector of 1s, and $\gamma$ is a tuning parameter.

In principle $\gamma$ can take any real value, however in practice the narrower range $-1 < \gamma < 1$ is sufficient for distinguishing nodes' positions [7, 11–13]. The tuning parameter $\gamma$ controls the interpretation of GC in the same way that $\beta$ controls the interpretation of BC. When $\gamma = 0$, GC is equal to degree. When $\gamma > 0$, GC assigns higher scores to nodes that are connected to well-connected neighbors, Finally, when $\gamma < 0$, GC assigns higher scores to nodes that are connected to poorly-connected neighbors. Accordingly, GC is conceptually similar to D1 when $\gamma < 0$.

Computing GC does not require a specialized software package. In R, it can be computed using:

```
G  <-  0  #Set  value  of  gamma
O  <-  matrix(1,  nrow  =  nrow(A))  #Define  vector  of  ones
GC  <-  A  %*%  ((A  %*%  O)^G)  #Compute
```

## Methods

### Harmonizing tuning parameters

One challenge comparing these tunable centrality measures is that their tuning parameters—$\alpha$, $\beta$, and $\gamma$—each have different scales. Rather than directly specify the value of each measure's tuning parameter, I instead define a meta-parameter $p$, and use it to define the tuning parameters according to a set of harmonizing functions.

First, for D1:

$$\alpha = p* - 1 \tag{4}$$

Eq 4 simply reverse-scores the $\alpha$ parameter.

Second, for BC:

$$\beta = \left(\frac{2}{e^{-p}} - 1\right) \times \frac{1}{\lambda_1} \tag{5}$$

Eq 5 applies the logistic function so that as $p$ approaches $\infty$, $\beta$ approaches its maximum allowable value, while as $p$ approaches $-\infty$, $\beta$ approaches its minimum allowable value.

Finally, for GC:

$$\gamma = p \tag{6}$$

Eq 6 uses $p$ as the value of $\gamma$.

Collectively, these harmonizing functions ensure that for all centrality measures, when $p > 0$ the measure functions as a centrality index by assigning higher scores to nodes that are connected to well-connected neighbors, while when $p < 0$ the measure functions as a power index by assigning higher scores to nodes that are connected to poorly-connected neighbors. In principle, $p$ can take any real value, however in the analyses below I consider only $-1 \leq p \leq 1$ because it is within this range that all three measures perform as identical centrality and power indices.

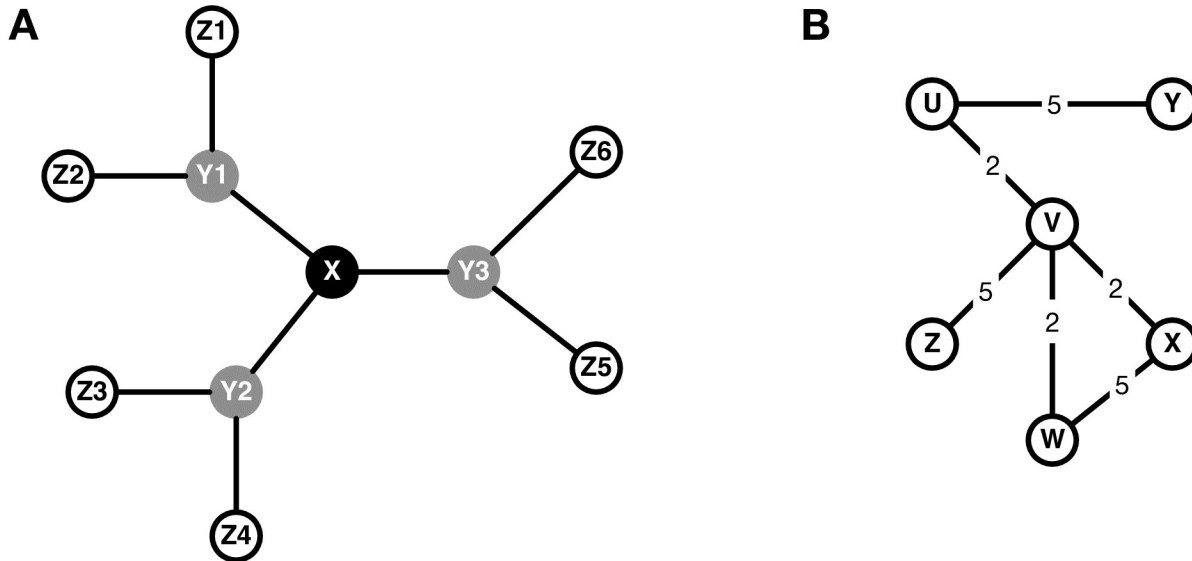

**Fig 1. (A) Unweighted [8] and (B) weighted [1] toy networks.**

## Networks

To evaluate the similarities between D1, BC, and GC, I compare them in five networks. First, I compare them in the toy network (see Fig 1A) from the exchange experiments [8] that initially led to the development of BC [6], and that has been used to validate BC and GC [6, 11]. Then I compare them in four additional networks used by Fronzetti Colladon and Naldi in their original analysis of D1: a weighted toy network (see Fig 1B), the unweighted Florentine marriage network [14], the weighted Zachary karate club network [15], and 1000 weighted scale-free networks. The scale-free networks were generated via preferential attachment. They contain 50 nodes with 2 edges added in each step, and the edges' weights are randomly assigned from the uniform distribution ranging from 1 to 20.

## Analysis

Within each network, I compute the vector of centrality scores using each measure for tuning meta-parameters in the range $-1 < p < 1$. I use the `distinctiveness` package for R to compute D1 [10], and use the R code shown above to compute BC and GC. Following [1], I then compute the Spearman correlation between D1 and BC, and between D1 and GC, for each $p$ to measure the similarity in node rankings that these measures yield. The code and data necessary to reproduce these results is available at https://osf.io/s6pqg/.

## Results

Fig 2 shows, for different values of the tuning meta-parameter, each position's distinctiveness centrality (Panel A), beta centrality (Panel B), and gamma centrality (Panel C), respectively, in the unweighted toy network shown in Fig 1A. When the tuning parameter is positive, these measures function as centrality indices and assign the highest score to position X because it is connected to nodes in position Y, which are well-connected. In contrast, when the tuning parameter is negative, these measures function as power indices and assign the highest score to position Y because it is connected to nodes in position Z, which are poorly-connected. Fig 2D

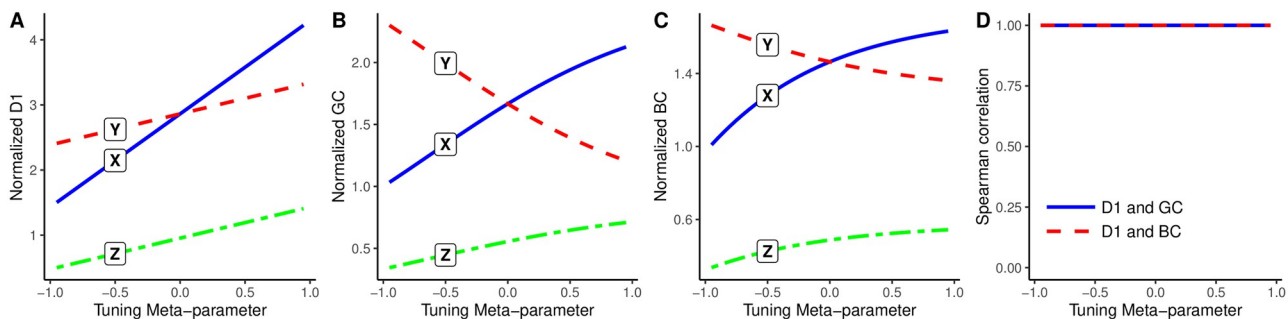

**Fig 2. (A) Distinctiveness centrality, (B) Beta centrality, (C) Gamma centrality, and (D) their Spearman correlations in an unweighted toy network.**

shows the Spearman correlation between D1 and BC scores (dashed red line), and between D1 and GC scores (solid blue line), at different values of the tuning meta-parameter $p$. It indicates that, for all tuning parameter values, these three measures yield identical node rankings in this network.

Fig 3 shows, for different values of the tuning meta-parameter, each position's distinctiveness centrality (Panel A), beta centrality (Panel B), and gamma centrality (Panel C), respectively, in the weighted toy network shown in Fig 1B. When the tuning parameter is positive and these measures function as centrality indices, position U is assigned a relatively low score (ranked 4th) because it is strongly connected to a neighbor that is not well-connected neighbor (Y). In contrast, position Z is assigned a relatively high score (ranked 3rd) because it is strongly connected to a well-connected neighbor (V). When the tuning parameter is negative and these measures function as power indices, position U is assigned a relatively high score (ranked 2nd) because it is strongly connected to a poorly-connected neighbor (Y). In contrast, position Z is assigned a relatively low score (ranked last) because it is strongly connected to a neighbor that is not poorly-connected (V). Fig 3D shows the Spearman correlation between D1 and BC scores (dashed red line), and between D1 and GC scores (solid blue line), at different values of the tuning meta-parameter $p$. It indicates that, for nearly all tuning parameter values, these three measures yield identical node rankings in this network.

Fig 4 shows the Spearman correlation between distinctiveness centrality and beta centrality (dashed red line), and between distinctiveness centrality and gamma centrality (solid blue line), for different values of the tuning meta-parameter $p$ in (A) the Florentine marriage

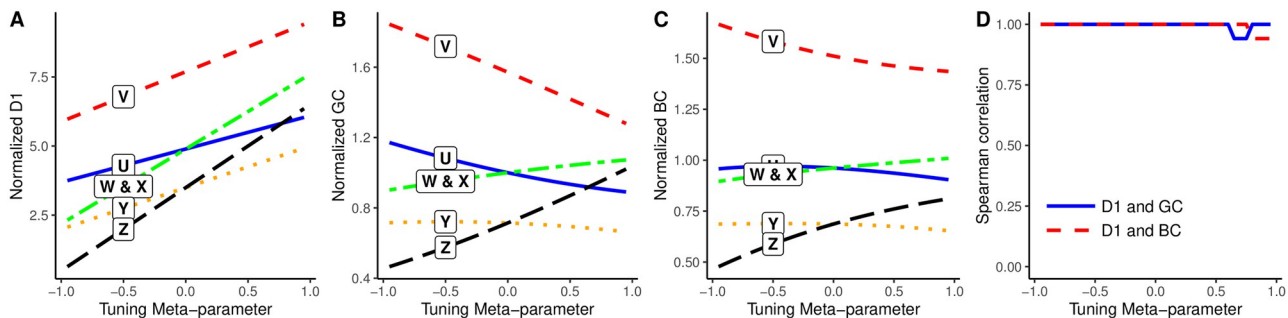

**Fig 3. (A) Distinctiveness centrality, (B) Beta centrality, (C) Gamma centrality, and (D) their Spearman correlations in a weighted toy network.**

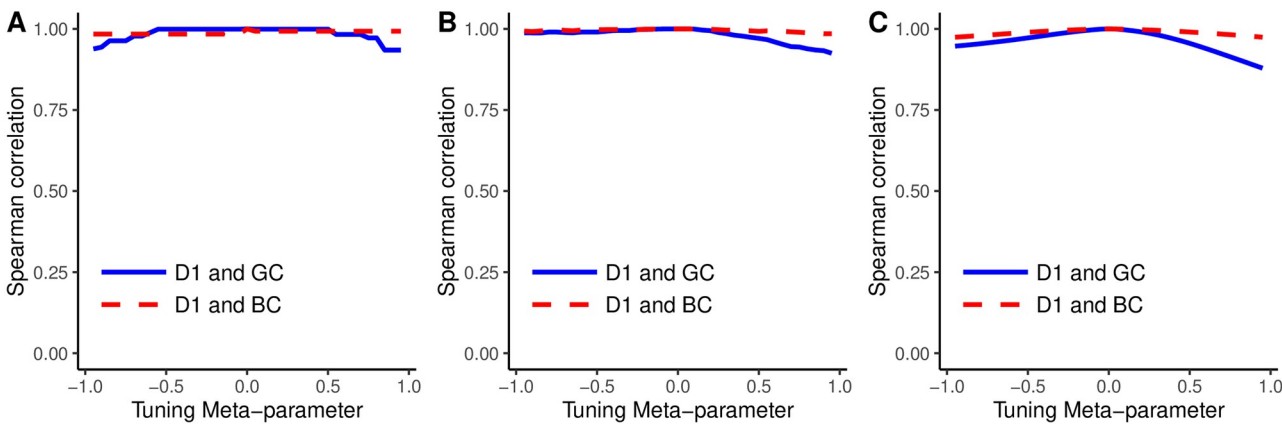

**Fig 4. Spearman correlations of distinctiveness centrality, beta centrality, and gamma centrality in (A) the Florentine marriage network, (B) Zachary karate network, and (C) 1000 weighted scale-free networks.**

network, (B) Zachary karate network, and (C) averaged over 1000 weighted scale-free networks. The uniformly high values of the Spearman correlation indicate that in all three networks and for all values of $p$, these three measures yield identical or nearly identical node rankings.

## Discussion

Believing that no existing centrality measures could identify nodes that are connected to poorly-connected neighbors, Fronzetti Colladon and Naldi [1] proposed a family of centrality measures called 'distinctiveness centrality' that were designed to fill this apparent gap. However, seemingly unbeknownst to Fronzetti Colladon and Naldi, two centrality measures already existed that identify nodes that are connected to poorly-connected neighbors: beta centrality first described in 1987 [6] and gamma centrality first described in 2011 [7, 12]. By comparing beta and gamma centrality to distinctiveness centrality in the same toy, empirical, and generated networks that Fronzetti Colladon and Naldi used in their original analysis, I have demonstrated that for plausible values of their respective tuning parameters these three measures yield identical or nearly identical rankings of nodes. These results indicate that distinctiveness centrality is not a new centrality measure, but instead is simply a minor variation of two centrality measures that already existed.

The fact that distinctiveness, beta, and gamma centrality yield identical or nearly identical node rankings under most circumstances means that in practice a researcher could use any of these measures and reach the same conclusions. However, there are several reasons that beta or gamma centrality should be preferred. First, beta and gamma centrality are already well-known in literature, where they are described in highly-cited articles and have been subsequently applied in articles that are themselves highly-cited. Second, beta and gamma centrality are flexible metrics that can (depending on the value of their tuning parameters) function as an index of centrality or an index of power. Third, beta and gamma centrality can be written elegantly in a matrix form that expresses the complete vector of centrality scores as a function of an adjacency matrix.

Therefore, researchers seeking to identify nodes that are connected to poorly-connected others should not use distinctiveness centrality. Instead, they should use either beta or gamma centrality because they are more widely-known in the literature, are more flexible, and are

computationally simpler. Additionally, researchers should be cautious when proposing new centrality measures, taking care to avoid duplicating measures that already exist.

## Author Contributions

**Conceptualization:** Zachary P. Neal.

**Data curation:** Zachary P. Neal.

**Formal analysis:** Zachary P. Neal.

**Methodology:** Zachary P. Neal.

**Software:** Zachary P. Neal.

**Visualization:** Zachary P. Neal.

**Writing – original draft:** Zachary P. Neal.

**Writing – review & editing:** Zachary P. Neal.

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
