## [Decision Letter · Decision Letter 0]

19 Dec 2023

PONE-D-23-34468Is `distinctiveness centrality' actually distinctive? A comment on Fronzetti Colladon and Naldi (2020)PLOS ONE

Dear Dr. Neal,

Thank you for submitting your manuscript to PLOS ONE. After careful consideration, we feel that it has merit but does not fully meet PLOS ONE’s publication criteria as it currently stands. Therefore, we invite you to submit a revised version of the manuscript that addresses the points raised during the review process.

We look forward to receiving your revised manuscript.

Kind regards,

Pilwon Kim

Academic Editor

PLOS ONE

Journal Requirements:

Reviewers' comments:

Reviewer's Responses to Questions

**Comments to the Author**

1. Is the manuscript technically sound, and do the data support the conclusions?

Reviewer #1: Partly

Reviewer #2: Yes

2. Has the statistical analysis been performed appropriately and rigorously? 

Reviewer #1: Yes

Reviewer #2: Yes

3. Have the authors made all data underlying the findings in their manuscript fully available?

Reviewer #1: Yes

Reviewer #2: Yes

4. Is the manuscript presented in an intelligible fashion and written in standard English?

Reviewer #1: Yes

Reviewer #2: Yes

5. Review Comments to the Author

Reviewer #1: The paper highlights an interesting point that the new centrality measures should be proposed carefully and only if they are required. They also suggest that it is better to use gamma or beta centrality measures than using the Distinctiveness centrality.

I have the following concerns that should be addressed by the author:

1. The paper lacks some relevant references as centrality measures have a vast literature. For example: the survey paper on this topic is also not cited -- Centrality measures in complex networks: A survey, by A. Saxena and S. Iyengar

2. At line 65, the author can mention how Fronzetti Colladon and Naldi chose the value of alpha from a wide range for the sake of completeness as now the reader has to go to the original paper to understand it better.

3. At line 123, in Equ. 4: author proposes to compute alpha based on p. However, it will make alpha vary from -infinity to +infinity and that does not make sense as mentioned by the author earlier. So please add an explanation for that. Please add how such a situation will be handled or controlled.

4. One main concern is that the results are shown on very small networks or BA networks that are not a good representation of real-world networks having communities. I will suggest adding results for large-scale networks.

5. The authors should also perform experiments on synthetic networks having modular/community structure to verify the applicability.

6. The results show the correlation of centrality measure value. However the centrality ranking is different from the centrality measures. In centrality ranking the nodes are ranked based on their centrality value. The author also should study the correlation of centrality ranking.

Reviewer #2: By theoretical and numerical analysis, this paper elaborates that the proposed 'distinctiveness centrality' for node importance is essentially the same as previous one. I think that the reason provided in the paper is sufficient, and the paper is also well organized. Therefore, I would like to recommend that this paper can be accepted in current version.

6. PLOS authors have the option to publish the peer review history of their article (what does this mean?). If published, this will include your full peer review and any attached files.

Reviewer #1: No

Reviewer #2: No

---

## [Decision Letter · Decision Letter 1]

30 Jan 2024

Is 'distinctiveness centrality' actually distinctive? A comment on Fronzetti Colladon and Naldi (2020)

PONE-D-23-34468R1

Dear Dr. Neal,

We’re pleased to inform you that your manuscript has been judged scientifically suitable for publication and will be formally accepted for publication once it meets all outstanding technical requirements.

Kind regards,

Pilwon Kim

Academic Editor

PLOS ONE

Additional Editor Comments (optional):

Reviewers' comments:

Reviewer's Responses to Questions

**Comments to the Author**

1. If the authors have adequately addressed your comments raised in a previous round of review and you feel that this manuscript is now acceptable for publication, you may indicate that here to bypass the “Comments to the Author” section, enter your conflict of interest statement in the “Confidential to Editor” section, and submit your "Accept" recommendation.

Reviewer #1: All comments have been addressed

2. Is the manuscript technically sound, and do the data support the conclusions?

Reviewer #1: Yes

3. Has the statistical analysis been performed appropriately and rigorously? 

Reviewer #1: Yes

4. Have the authors made all data underlying the findings in their manuscript fully available?

Reviewer #1: Yes

5. Is the manuscript presented in an intelligible fashion and written in standard English?

Reviewer #1: Yes

6. Review Comments to the Author

Reviewer #1: The authors have addressed all my comments. Therefore I recommend paper for the publication. I do not have any further concerns.

7. PLOS authors have the option to publish the peer review history of their article (what does this mean?). If published, this will include your full peer review and any attached files.

Reviewer #1: **Yes: **Akrati Saxena

---

## [Editor Report · Acceptance letter]

20 Feb 2024

PONE-D-23-34468R1 

PLOS ONE

Dear Dr. Neal, 

I'm pleased to inform you that your manuscript has been deemed suitable for publication in PLOS ONE. Congratulations! Your manuscript is now being handed over to our production team.

Kind regards, 

on behalf of

Professor Pilwon Kim 

Academic Editor

PLOS ONE